

# Effect of sleep quality on vision-related quality of life in a sample of Qassim University students: an ocular surface health approach

Hanan Awad Alkozi and Saleh Alshammeri

Department of Optometry, College of Applied Medical Sciences, Qassim University, Buraydah, Saudi Arabia

## ABSTRACT

**Background:** Optimal sleep is essential for maintaining physiological homeostasis, and its disruption is associated with numerous physical and mental health issues. Previous studies have linked poor sleep quality to ocular conditions such as dry eye. However, the relationship between sleep quality and vision-related quality of life remains underexplored. This study aimed to investigate the association between ocular surface health and sleep quality, and how both factors relate to vision-related quality of life.

**Methods:** Ethical approval was granted by the Deanship of Graduate Studies and Scientific Research at Qassim University (reference no. 21-20-09), and the study was conducted in accordance with the Declaration of Helsinki. Young adult participants completed three questionnaires: the Pittsburgh Sleep Quality Index, the National Eye Institute Visual Function Questionnaire, and the Ocular Surface Disease Index. Ocular surface assessments were conducted using the Keratograph 5M and the Schirmer test to measure tear production.

**Results:** Analysis revealed significant correlations between ocular symptoms and vision-related quality of life (Spearman's $r = 0.47$, $p < 0.0001$, $n = 93$), sleep efficiency ($r = 0.22$, $p < 0.05$, $n = 75$), and sleep disturbances ($r = 0.25$, $p < 0.05$, $n = 75$). A significant negative correlation was also found between sleep quality and vision-related quality of life ($r = -0.31$, $p < 0.005$, $n = 75$). These findings highlight the impact of sleep quality on ocular health, particularly the ocular surface.

**Conclusion:** The results suggest a meaningful link between sleep quality, ocular surface health, and vision-related quality of life. Further research is needed to clarify the causal mechanisms and risk factors contributing to ocular discomfort and its effects on visual well-being.

# INTRODUCTION

Sleep, accounting for approximately one-third of our lives, is essential for maintaining vital physiological functions (*Colten & Altevogt, 2006*). The World Health Organization (WHO) has declared reduced sleep duration a public health concern (*Chattu et al., 2019*).

Corresponding author
Hanan Awad Alkozi,
h.alkozi@qu.edu.sa

Both extremes, from insufficient to excessive sleep, are related to numerous cognitive and physical comorbidities. Poor sleep quality is characterized by different factors, such as increased sleep latency, disturbances, and increased movement while sleeping. Evidence indicated a strong association between sleep quality and various chronic diseases, such as hypertension, diabetes, cardiovascular diseases, and even cancer (*Berisha, Shutkind & Borniger, 2022*; *Lee & Lee, 2022*; *Miller & Howarth, 2023*)

The harmful influence of sleep disorders on the quality of life related to health has been previously emphasized, highlighting the burden of prevalent poor sleep quality in modern societies (*Zeidler et al., 2018*). Quality of life is multifaceted, incorporating physical, social, and psychological domains, all of which are associated with and affected by sleep quality. Furthermore, sleep disorders can exacerbate existing medical conditions. For example, insomnia and daytime sleepiness can negatively impact general and mental health, though not consistently in the physical domain. While some studies have shown no impact of sleep apnea, restless leg syndrome, and snoring on overall health-related quality of life (*Chen, Gelaye & Williams, 2014*). This finding is not universal since controversial results were found between studies (*Wanberg et al., 2021*). Notably, a Japanese study of over 9,000 participants found that sleep quality and physical/mental quality of life were associated independently of sleep duration (*Matsui et al., 2021*). Another study done in Jordan on 1,422 young adults has shown that poor sleep quality is associated with higher levels of both anxiety and depression (*Alkhatatbeh, Abdul-Razzak & Khwaileh, 2021*). To subjectively evaluate the quality of sleep, the Pittsburgh Sleep Quality Index (PSQI) questionnaire is widely used due to its high validity and ease of use. It comprises 19 items, and the calculated results with higher scores indicate poorer sleep quality (*Carpenter & Andrykowski, 1998*).

Among the negative health outcomes affected by poor sleep quality is the high occurrence of dry eye disease. Research has established a link connecting sleep quality and both signs as well as symptoms of dry eye syndrome (*He et al., 2022*; *Ayoubi et al., 2024*). Notwithstanding, the specific association between sleep quality and quality of life related to vision requires further investigation, as no prior studies exploring this link were identified.

Vision plays a critical role in our daily activities, like driving, watching television, reading, and facial recognition. Moreover, visual impairment is a significant concern for public health, ranking as the ninth most prevalent disability in the United States, and its influence on overall life quality has been extensively studied (*Vannasdale et al., 2021*). To appraise the quality of life specifically related to vision, the National Eye Institute developed the Visual Function Questionnaire (NEI-VFQ). Initially consisting of 51 items (*Mangione et al., 1998*), followed by the development of a shortened 25-item version (NEI-VFQ25), which demonstrated strong reliability and accuracy. The NEI-VFQ25 is now a commonly implemented instrument to assess vision-related life quality (*Mangione et al., 2001*; *Nickels et al., 2017*). This questionnaire encompasses 12 domains: general health, general vision, near and distance activities, driving, peripheral vision, color vision, ocular pain, vision-related role difficulties, dependency, social function, and mental health (*Mangione et al., 2001*). Several studies aiming to evaluate quality of life in individuals with

ocular conditions, including uveitis, glaucoma, retinitis pigmentosa, and dry eye disease, have made use of the NEI-VFQ25 (*Nichols, Mitchell & Zadnik, 2002*; *Sainohira et al., 2018*; *Rossi et al., 2022*).

Dry eye disease (DED) is a multifaceted condition affecting the ocular surface and tear film (*Tsubota et al., 2020*). Common symptoms include foreign body sensation, burning, itching, and reduced visual acuity. DED presents a substantial burden, particularly with its rising prevalence in the past years (*Willcox et al., 2017*).

DED is a frequent complaint in eye care clinics, with prevalence estimates spanning from 5% to 50% (*Alkhaldi et al., 2023*; *Papas, 2021*). The disruption of tear film homeostasis in DED can be caused by various factors, such as age, gender, and certain medications (*Craig et al., 2017*; *Gomes et al., 2023*). Prior research has indicated a relationship between sleep quality and DED, highlighting the detrimental effects of dry eye syndrome on general health and quality of life related to vision (*Li et al., 2012*; *Erøy, Utheim & Sundling, 2023*). In this sense, several validated questionnaires to assess dey eye symptoms are available and widely used; The Ocular Surface Disease Index (OSDI) is among the most used self-reporting methods to measure the severity of dry eye symptoms due to its rapid diagnosis of ocular surface disease (*Aljarousha et al., 2024*; *Schiffman et al., 2000*).

The rising prevalence of dry eye disease in the Middle East has become pronounced, according to a recent systematic review and meta-analysis, which has highlighted a significant regional variation in dry eye disease; the results showed that dry eye prevalence in the Middle East was higher than the global estimate (*Mohamed et al., 2024*).

This study aims to explore the association connecting self-reported sleep quality and vision-related quality of life, specifically concerning the signs and symptoms of ocular surface disorders.

# MATERIALS AND METHODS

## Participants

The *G*Power* software version 3.1.9.7 is used for sample size determination. The main statistical analysis to be conducted was linear correlation analysis between PSQI questionnaire, vision-related quality of life and research variables. The parameters used to determine sample size were as follows: Two tailed tests, average effect size assumed, 95% confidence level, power $(1−\beta)$ 0.80. Based on the assumptions mentioned, the minimum required sample size was 84 students. A total of 93 university students were enlisted for this study through convenient sampling. All participants were offered information regarding the research and signed a written informed consent form. The research was conducted following the Helsinki Declaration and has been approved by the Deanship of Graduate Studies and Scientific Research at Qassim University (reference no. 21-20-09). Excluded participants either received sleep aid medication, ocular surgery, or presented with an ocular disease other than dry eye. All subjects recruited had never experienced any of the exclusion criteria outlined in the current study.

## Ocular examination

Ocular surface health was evaluated through a series of examinations. All participants underwent slit-lamp biomicroscopy to assess limbal health, such as neovascularization or signs of pterygium. And the eyelids to look for signs of blepharitis or abnormal lash growth or lid abnormalities. Slit lamp examination was followed by measurements using the Keratograph 5M. This device, an ocular surface imaging system, enables non-invasive assessment of various parameters, including, conjunctival redness, inferior tear meniscus height (TMH), lipid layer analysis *via* interferometry, non-invasive tear breakup time (NIKBUT), and meibography for evaluating the meibomian glands of the lower and upper eyelids after eversion (*Alfaro-Juárez et al., 2019*). Tear quantity was subsequently assessed using the Schirmer I test. This test involves placing a commercially available Schirmer strip at the temporal intersection of the lower eyelid margin, instructing the patient to close their eyes for 5 min, and then measuring the length of the moistened portion of the strip, which indicates the volume of tears absorbed. A Schirmer test result of less than 10 mm of moisture is suggestive of reduced tear production and is considered a potential indicator of dry eye syndrome (*Willcox et al., 2017*). All patients went through all examinations between 10:30 a.m. and 2:30 p.m. to ensure no diurnal variability, and all the following tests and surveys were conducted at the same time.

## Ocular surface disease index

Participants were provided a link to an online survey hosted on Google Forms, which included the three study questionnaires. The Ocular Surface Disease Index (OSDI) was utilized to evaluate dry eye symptoms. This Arabic version of the OSDI, utilized in the study, was translated and verified by *Bakkar, El-Sharif & Qadire (2021)*, which Dr. May Bakkar generously provided. As a self-administered instrument, the OSDI yields scores that categorize the acuteness of symptoms for dry eye. Scores are interpreted as follows: normal (0–12), mild (13–22), moderate (23–32), and severe (above 33). Scoring procedures adhered to the manufacturer's guidelines (*Okumura et al., 2020*).

## Sleep quality assessment

The Arabic variation of the Pittsburgh Sleep Quality Index (PSQI) was utilized to evaluate sleep quality, a well-established and validated self-report measure (*Suleiman et al., 2010*). This instrument is widely recognized for effortless administration and robust psychometric traits, including high validity, making it a valuable tool in sleep research (*Buysse et al., 1989*). The PSQI comprises 19 items that assess seven distinct dimensions of sleep: subjective sleep quality, sleep latency, sleep duration, habitual sleep efficiency, sleep disturbances, use of sleep medication, and daytime dysfunction. Each dimension's score ranges from 0 to 3, with lower scores indicating better quality of sleep. Spanning from 0 to 21, a global PSQI score is determined by a summation of the individual component scores. A score of five or greater on the global PSQI is commonly used to define poor sleep quality (*Buysse et al., 1989*).

## Quality of life related to vision

Quality of life related to vision was determined using NEI-VFQ-25 self-report questionnaire (*Nichols, Mitchell & Zadnik, 2002*). The questionnaire used was an Arabic validated version (*Abdelfattah et al., 2014*). Each scale of the test was graded from 0 to 100, scoring method followed the protocol in the manual offered by the National Eye Institute. Each test subscale was graded from 0 to 100 where lower scores represent poor function, then the overall score was summed, with higher scores denoting better quality of life related to vision (*Mangione et al., 2001*).

## Data analysis

Statistical analysis was done using Prism GraphPad version 10 (San Diego, CA, USA). Descriptive analysis was done to describe demographics and clinical data. The normality of the data obtained was tested using the Shapiro-Wilk test. The results did not pass the normality test, as a result, Spearman R non-parametric test for correlation was used for statistical analysis.

# RESULTS

## Study population

The research consisted of 93 university students from Qassim University in Qassim Province, Saudi Arabia. The mean age was $22.25 \pm 1.38$ years old, of which 63.82% were males. All subjects involved had never been diagnosed with ocular diseases, never had ocular surgery, nor used sleep aid.

## Dry eye-signs and symptoms

To establish a sound correlation between dry eye and sleep quality and quality of life related to vision, it was necessary to first confirm the relationship between dry eye signs and symptoms. Based on several previous studies, dry eye symptoms do not always align with its signs (*Bartlett et al., 2015*). The sample recruited in the current study showed that only tear breakup time correlated negatively with scores obtained from OSDI questionnaire. Patients with lower NIKBUT had higher OSDI scores that indicated worse dry eye symptoms with Spearman r of $-0.233$ and $-0.253$ for right and left eyes respectively ($p < 0.05$, $n = 93$). Other measurements taken, such as tear meniscus height, Schirmer test, meibography, conjunctival redness, and lipid interferometry, did not reveal a significant correlation with dry eye symptom scores (please refer to Supplemental Data for further information).

## Quality of life related to vision and dry eye

The data analysis obtained by clinical examination using the keratograph 5M instrument together with the Schirmer test measurements to find a possible correlation between dry eye signs with quality of life related to vision, results showed only a significant correlation between NIKBUT and one domain of quality of life related to vision: vision-related role

**Table 1 VFQ-NIE- 25 Spearman correlation with dry eye signs and symptoms.**

| | Quality of life | General health | General vision | Ocular pain | Near activities | Distance activities | Social functioning | Mental health | Role difficulties | Dependancy | Driving | Color vision | Peripheral vision |
|---|---|---|---|---|---|---|---|---|---|---|---|---|---|
| OSDI | −0.47**** | −0.13 | −0.34*** | −0.49**** | −0.30** | −0.40*** | −0.20 | −0.23* | −0.27* | −0.26* | −0.35** | −0.04 | −0.18 |
| Schrimer test (OD) | 0.16 | 0.05 | 0.20 | −0.02 | 0.09 | 0.12 | 0.07 | 0.16 | 0.04 | 0.19 | 0.12 | −0.11 | 0.13 |
| Schrimer test (OS) | −0.02 | −0.09 | −0.06 | −0.13 | −0.07 | −0.07 | −0.04 | −0.01 | −0.26 | −0.11 | −0.10 | −0.02 | 0.06 |
| TMH (OD) | 0.18 | 0.02 | −0.06 | 0.14 | −0.05 | 0.06 | −0.03 | 0.09 | −0.06 | 0.05 | 0.21 | 0.08 | 0.04 |
| TMH (OS) | 0.22 | 0.08 | 0.01 | 0.05 | −0.07 | 0.10 | 0.08 | 0.21 | −0.14 | 0.10 | 0.18 | 0.11 | 0.17 |
| NIKBUT (OD) | 0.20 | −0.10 | −0.22 | −0.03 | 0.07 | 0.09 | 0.04 | 0.01 | 0.21 | 0.06 | 0.41*** | 0.13 | 0.20 |
| NIKBUT (OS) | 0.23 | 0.02 | −0.06 | 0.01 | 0.05 | 0.06 | −0.03 | 0.06 | 0.10 | 0.00 | 0.34** | 0.18 | 0.37 |
| LLT (OD) | 0.03 | −0.04 | −0.03 | −0.12 | −0.10 | −0.11 | −0.11 | −0.10 | −0.20 | −0.07 | −0.05 | 0.10 | 0.20 |
| LLT (OS) | 0.16 | −0.09 | −0.09 | 0.00 | 0.03 | −0.06 | −0.04 | −0.10 | −0.21 | −0.12 | −0.01 | −0.01 | 0.08 |
| Meibography (OD) | −0.20 | −0.18 | −0.10 | −0.24* | −0.29* | −0.13 | −0.22 | 0.00 | −0.11 | 0.10 | −0.04 | −0.12 | −0.14 |
| Meibography (OS) | −0.16 | −0.13 | −0.17 | −0.24* | −0.18 | −0.08 | −0.17 | −0.22 | −0.17 | 0.04 | −0.01 | −0.06 | −0.05 |

**Notes:**

The table represents the Spearmens r value of the mentioned ocular signs and symptoms of dry eye *vs.* all parameters studied in the vision-related quality of life questionnaire: OSDI, Ocular Surface Disease Index. TMH, Tear Meniscus Height; NIKBUT, Non-invasive Tear Break-up Time; LLT, Lipid Layer Thickness; bold, positive correlation; italic, negative correlation. Only pronounced colors with asterisks are statistically significant correlations.

**** $p < 0.0001$.
*** $p < 0.001$.
** $p < 0.01$.
* $p < 0.05$.

difficulty; with Spearman r scores being for the and right eye = 0.41, left eye = 0.34 ($p < 0.0001$, $p < 0.001$, respectively, $n = 93$). Also, Meibography scores showed a negative correlation with the ocular pain domain with Spearman's r of −0.24 for both eyes ($p < 0.05$, $n = 93$) (Table 1).

When data was analyzed to find a correlation between symptoms of dry eye measured by OSDI questionnaire and quality of life related to vision, the results showed a highly significant correlation linking the overall OSDI scores and the overall NEI-VFQ-25 (spearman r = 0.47, $p < 0.0001$, $n = 93$). When analyzing the OSDI score with each item in the NEI-VFQ-25, driving, peripheral vision, near and distance activities, color vision, vision-related role difficulty, ocular pain, social function, dependency, and mental health, the result showed significant correlation in most domains (Table 1). As expected, OSDI scores were not associated with either general health, social functioning, color vision, or peripheral vision domain of the NIE-VFQ-25 questionnaire.

## Quality of life related to vision and sleep quality metrics

Before analyzing the data, some participants' answers were excluded from the analysis due to lack of completion of the questionnaire, the completed questionnaire rate was 80.6%, resulting in a sample number of 75 participants. Of all the analyzed data, 52% had poor sleep quality scores (with a PSQI sum of more than 5).

**Table 2  Correlation between vision-related quality of life and sleep quality parameters.**

|  | Vision-related quality of life (Spearman r value) | P value |
|---|---|---|
| Subjective sleep quality | −0.23 | 0.040* |
| Sleep onset latency | −0.05 | 0.690 |
| Sleep duration | −0.18 | 0.116 |
| Habitual sleep efficiency | −0.31 | 0.005** |
| Sleep disturbances | −0.34 | 0.002* |
| Use of sleep medication | −0.13 | 0.268 |
| Daytime dysfunction | −0.36 | 0.001** |
| Overall PSQI | −0.31 | 0.005** |

Notes:
** $p < 0.01$.
* $p < 0.05$.

**Table 3  Correlation between dry eye symptoms and sleep quality components.**

|  | OSDI (Spearman r value) | P value |
|---|---|---|
| Subjective sleep quality | 0.04 | 0.72 |
| Sleep onset latency | 0.11 | 0.34 |
| Sleep duration | −0.06 | 0.61 |
| Habitual sleep efficiency | 0.22 | 0.04* |
| Sleep disturbances | 0.25 | 0.03* |
| Use of sleep medication | 0.03 | 0.77 |
| Daytime dysfunction | 0.10 | 0.38 |
| Overall PSQI | 0.15 | 0.21 |

Note:
* $p < 0.05$.

The total NEI-VFQ-25 score obtained was analyzed for correlation with each component of the PSQI questionnaire, and a high negative correlation was found between the overall score NEI-VFQ-25 and PSQI ($r = −0.31$, $p < 0.005$, $n = 75$). Vision-related life quality was revealed to correlate significantly with subjective quality of sleep ($r = −0.23$, $p < 0.05$, $n = 75$), habitual sleep efficiency ($r = −0.31$, $p < 0.005$, $n = 75$), and disturbances in sleep ($r = −0.34$, $p < 0.005$, $n = 75$). All results are shown in Table 2.

### Dry eye and sleep quality

Ultimately, ocular symptoms were associated with quality of sleep metrics but not the signs of dry eye. The statistically significant relation with dry eye was noted between OSDI and sleep efficiency ($r = 0.22$, $p < 0.05$, $n = 75$), and sleep disturbances ($r = 0.25$, $p < 0.05$, $n = 75$), all components are shown in Table 3.

## DISCUSSION

This study investigated the relationships between dry eye signs and symptoms, quality of life related to vision, and sleep quality. The results indicated a significant association between ocular symptom severity, as measured by the Ocular Surface Disease Index (OSDI) questionnaire, and quality of life related to vision, sleep efficiency, and sleep

disturbances. Furthermore, the established association between sleep quality and quality of life related to vision was corroborated. These findings present both similarities and differences when compared to previous research. Published data shows that sleep problems are becoming an increasing problem globally (*Lund et al., 2010*). For instance, a study done in Australia revealed that between 41% and 42% of young adults with a mean age of 22.2 years suffered from sleep issues, with males scoring significantly higher than females (*McArdle et al., 2020*). University students usually experience numerous factors to cope with and keep with their academic responsibilities, and often they suffer from several factors affecting their quality of life, sleep is among the reported factors (*Schlarb, Friedrich & Claßen, 2017*). Higher percentages were reported in a study population of university students in Jordan where sleep quality was found to be poor in 74% of a total of 1,308 subjects (*Albqoor & Shaheen, 2021*). A similar study with the same population group (University students) was done in Germany and revealed that 48.7% suffered from poor sleep quality (*Schmickler et al., 2023*). In the current sample, poor sleep quality obtained by PSQI scores higher than five accounted for 52%.

Dry eye symptoms are not necessarily reflected by the signs seen in the clinic; therefore, numerous questionnaires were developed to self-report dry eye symptoms. One of the most used questionnaires is the Ocular Surface Disease Index (OSDI), specifically designed for dry eye patients to assess dry eye-related symptoms, functional limitations, and environmental factors (*Dana et al., 2019*). Aligned with the results obtained here, only the NIKBUT test was related with increasingly poor dry eye symptoms. Moreover, results in the present work showed an association with dry eye symptoms and only two components of the sleep quality questionnaire: sleep efficiency and sleep disturbances. These results are in part contradictory to previous work where OSDI scores correlated with all PSQI components except the use of sleep medication (*Ayoubi et al., 2024*). These differences could be due to the difference in the age group and the population studied since they examined veterans with a mean age of 56 ± 5 years, whereas our population mean age was 22.25 ± 1.38, which highlights the importance of testing different age groups due to plausible differences. Unlike the present study, another study showed that Schirmer test results were correlated with PSQI score, indicating that further studies are necessary to elucidate the question regarding dry eye signs and sleep quality (*Kawashima et al., 2016*).

Among the questions that authors wanted to clarify in the current manuscript was the association between sleep quality and quality of life related to vision. A question was raised because it is well established in the literature that dry eye has a negative influence on the quality of life, in addition to the fact that both measurements of interest are subjective and dependent on the sole answer of the participant. Moreover, only one study that correlated sleep quality and NEI-VFQ-25 and it was in a geriatric population, yet similar results were seen in the current study. Worse sleep quality scores are reported to be significantly associated with poor vision-related quality of life.

## CONCLUSIONS

This research highlights an association between sleep quality, quality of life related to vision, and dry eye disease. Nevertheless, the study has limitations, and still, more studies

are required to highlight the capacity of healthy and good sleep quality in promoting better ocular surface health. The population recruited in the study was healthy and young, which established a base for future studies. However, increasing the sample and dividing the subjects into categories with the degree of dry eye present would be beneficial. This was not done since only profound results were observed between the symptomatology of dry eye and the aspects of interest.

### Funding
The researchers were supported by the Deanship of Graduate Studies and Scientific Research at Qassim University (QU-APC-2025). The funders had no role in study design, data collection and analysis, decision to publish, or preparation of the manuscript.

### Grant Disclosures
The following grant information was disclosed by the authors:
Deanship of Graduate Studies and Scientific Research at Qassim University: QU-APC-2025.

### Competing Interests
The authors declare that they have no competing interests.

### Author Contributions
- Hanan Awad Alkozi conceived and designed the experiments, performed the experiments, analyzed the data, prepared figures and/or tables, authored or reviewed drafts of the article, and approved the final draft.
- Saleh Alshammeri performed the experiments, analyzed the data, authored or reviewed drafts of the article, and approved the final draft.

### Human Ethics
The following information was supplied relating to ethical approvals (*i.e.*, approving body and any reference numbers):

The ethical approval was granted from Deanship of Graduate Studies and Scientific Research at Qassim University (reference no. 21-20-09).

### Data Availability
The raw measurements are available in the Supplemental File.

### Supplemental Information
Supplemental information for this article can be found online at http://dx.doi.org/10.7717/peerj.19801#supplemental-information.

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
