# Peer review of "Effect of sleep quality on vision-related quality of life in a sample of Qassim University students: an ocular surface health approach"

_PeerJ, doi:10.7717/peerj.19801_

## Round 0.1 · original submission · Major Revisions

· Academic Editor

Major Revisions

·

Basic reporting

1. The title should be ‘‘Effect of Sleep Quality on Vision-Related Quality of Life in a Sample of Qassim University Students: An Ocular Surface Health Approach’’.
2. Lines 39–40, 61-63, 78, and 82–83: Please add appropriate citation(s).
3. Line 40: Please add the abbreviation (WHO) immediately after the first mention of World Health Organization.
4. Line 42: Please provide a classification system for sleep quality scores, including categories such as poor, average, and good.
5. Lines 51-55, 61-63, 122-124: The sentence structure rephrasing for better clarity.
6. Lines 56-57: Please add a recent study from the Middle East after the Japanese study.
7. Lines 60-61, 72, 77 and 86-87: Please rearrange the citations once again.
8. Line 72, Add a full stop.
9. Line 88-90: Please highlight the problem statement in the Middle East/Arab population before mentioning the research objective.
10. In the introduction, you mentioned the NEI-VFQ-25; however, you used the OSDI and PSQI in your study. Please highlight these questionnaires in the introduction.

Experimental design

1. Line 40: Please add the baseline characteristics table of the 93 university students, such as gender, level of education, area of study, year of study, and duration of study, etc.
2. For the sample size calculation, please add the formula (equation), the overall number of Qassim University students, and provide appropriate citation(s).
3. Which type of sampling method was used?
4. The exclusion criteria are unclear. Did the participants receive sleep aid medication, undergo ocular surgery, or present with an ocular disease other than dry eye within the past three months, six months, or one year? I recommend that you create a table outlining the inclusion and exclusion criteria.
5. Who conducted the clinical tests? Are they undergraduate students? If they are students, did you perform the validity of the results?
6. Were the questionnaires and clinical tests completed in the same session?
7. Lines 107–108: Please specify the types of clinical tests conducted using SL biomicroscopy.
8. What were the gradings criteria for TMH, MGDs, and NIKBUT in the current study?
9. Why didn’t you use a multivariate logistic regression model in the current study? I also recommend highlighting the statistical tests used in the analysis.
10. The method section is unclearly written in the abstract. Please rephrase it for better clarity.

Validity of the findings

1. I recommend that you create a table that presenting a mean (SD)/Median (IQR) for baseline characteristics, dry eye and quality of life questionnaires and clinical signs.
2. Line 167: Please add appropriate citation(s).
3. The symbols r, p, and n should be in italics and the abbreviations should be mentioned in Tables 2 and 3.
4. Table 1: May I know why you used different colors such as red, light green and dark green? Additionally, please include the p-values and specify that you selected p < 0.25 for inclusion in the multivariate logistic regression model.
5. Table 2: Please add the grades for sleep quality parameters such as sleep duration, sleep disturbances and use of sleep medication, etc. Additionally, you used the word ‘‘Overall’’ before the name of the questionnaire in the manuscript, but in Table 2, you used ‘‘Total’’ word before the PSQI.
6. The results are not well written; they should be clearer and more thoroughly explained.

Additional comments

1. Line 211, Lines 223-224: Please add appropriate citation(s).
2. From lines 213–222, you included previous studies from Australia and Germany. Please also include recent studies from the Middle East or an Arab country.
3. The discussion section is well written.
4. Ayoubi, Cabrera, Mangwani, Locatelli, & Galor, 2024; Ayoubi et al., 2024), (Buysse, Reynolds, Monk, Berman, & Kupfer, 1989; Buysse et al., 1989), (Nichols, Mitchell, & Zadnik, 2002; Nichols et al., 2002): you have written the same reference in different styles.

Reviewer 2 ·

Basic reporting

Dear Authors,
Thank you again for submitting your manuscript, "Sleep quality effect on vision-related quality of life: ocular surface health approach." I appreciate the opportunity to provide further feedback on your valuable work. My aim is to offer suggestions that may help strengthen and clarify certain aspects of the manuscript based on the review criteria and the provided sources.
Overall, the study addresses an important intersection of sleep, ocular health, and quality of life, and the methodology appears sound for this exploratory cross-sectional design.
1. Abstract Consistency (Results Section):There appears to be a discrepancy between the results summary provided in the Abstract and the detailed data presented in Table 3. The Abstract states: "Analysis revealed a significant correlation between ocular symptoms and: quality of life related to vision (Spearman's r = 0.47, p < 0.0001, n = 93), sleep efficiency, and sleep disturbances, respectively (r = -0.25, p < 0.05, n = 75)". However, Table 3 shows the correlation between OSDI (ocular symptoms) and Habitual sleep efficiency is r=0.09 with p=0.43, which is not statistically significant. The correlation between OSDI and Sleep disturbances is r=0.25 with p=0.03*, which is statistically significant. The r-value of -0.25 in the abstract for "sleep efficiency, and sleep disturbances, respectively" also seems to potentially combine two values or misrepresent the sign, as the significant correlation with Sleep Disturbances in Table 3 is positive (r=0.25). Please carefully review the results summary in the Abstract and align it precisely with the significant correlations found in Table 3. Ensure the r-values, p-values, and the sleep components stated as significant correlations with ocular symptoms match the table data.

Similar to the abstract, the narrative in the Results section states: "The statistically significant relation with dry eye was noted between OSDI and habitual sleep, and sleep disturbances (r=-0.25, p<0.05, n=75)". Again, Table 3 shows OSDI vs. Habitual sleep efficiency (r=0.09, p=0.43 - not significant) and OSDI vs. Sleep disturbances (r=0.25, p=0.03* - significant). The statement in the narrative incorrectly identifies "habitual sleep efficiency" as having a statistically significant relation with OSDI based on Table 3, and repeats the r-value of -0.25 which also seems inconsistent with the positive r=0.25 in the table. Please revise the narrative in the Results section to accurately reflect the findings presented in Table 3. Specifically, correct the statement regarding the correlation between OSDI and habitual sleep efficiency, and ensure the r-value and its sign for sleep disturbances are correctly reported as shown in Table 3.

While the manuscript is generally clear and professional, there are a few instances where phrasing could be slightly refined for improved flow and precision. The lines 64-67, 83-84, and 142 as areas for minor adjustments. For example, "Vision plays a critical role in our daily activities, like driving, watching television, reading, and facial recognition. Moreover, in public health, visual impairment is a significant concern..." could be phrased to connect the ideas more smoothly. "The disruption of tear film homeostasis in DED can be caused by various factors, such as age, gender, and certain medications" is clear, but a slight rephrase might enhance flow. "Quality of life related to vision was determined using NEI-VFQ-25 self-report questionnaire (Nichols et al., 2002). The questionnaires Arabic version was used (Abdelfattah et al., 2014)." could be slightly adjusted for sentence structure. I suggest a final polish of the language throughout the manuscript, perhaps focusing on transitions and sentence structure in places like those previously mentioned. This can help ensure maximum clarity for an international audience.

Experimental design

The manuscript states that validated Arabic versions of the OSDI and NEI-VFQ-25 questionnaires were used. While citations are provided for the original and Arabic versions, confirming the specific version of the PSQI used (e.g., acknowledging if it was a validated Arabic version or translated for this study with appropriate checks) would enhance detail. The manuscript mentions an "Arabic variation" and cites a study on translating the PSQI into Arabic, but doesn't explicitly state this version was validated for this population. Please explicitly state whether the Arabic version of the PSQI used in this study was a previously validated version (and cite it if so) or if it was translated or adapted specifically for this study. If adapted, mentioning any steps taken to ensure its reliability and validity in this population would be helpful for replication purposes.

Validity of the findings

The results show a significant correlation between OSDI and Sleep Disturbances (r=0.25, p=0.03*), and between Total PSQI and Total NEI-VFQ-25 (r=-0.34, p=0.005**). The discussion correctly points out that the finding of only one PSQI component correlating with OSDI differs from another study and suggests age as a potential factor. While appropriately discussed, emphasizing this specific finding (OSDI only correlating with Sleep Disturbances, not other PSQI components, in this young cohort) in contrast to findings in older populations is a key nuance of your results. You already do this well, but ensuring its prominence might be beneficial. It underscores the value of studying different age groups.

---

## Round 0.2 · accepted · Accept

· Academic Editor

Accept

Dear Authors

Your revision has been reviewed by two experts in the field, and we recognize that the quality of the manuscript has been improved. Your submission is now endorsed by two experts for acceptance of publication in PeerJ. One last thing to do before the publication. Please address the limitation as the number of participants dropped to 75 for some quaternaries (quality of life related to vision and sleep quality metrics)

Thank you for submitting your article to PeerJ. I would like to express my gratitude for your contributions and efforts to the scientific community. I look forward to receiving your research and review articles in the future.

Best Regards

Yung-Sheng Chen, Ph.D.
Academic Editor

·

Basic reporting

All the corrections based on my comments have been completed.

Experimental design

All the corrections based on my comments have been completed.

Validity of the findings

All the corrections based on my comments have been completed.

Reviewer 2 ·

Basic reporting

-

Experimental design

Addressed

Validity of the findings

Addressed

Additional comments

Addressed